# GIST: Improving Parameter Efficient Fine-Tuning via Knowledge Interaction

## ABSTRACT

Recently, the Parameter Efficient Fine-Tuning (PEFT) method, which adjusts or introduces fewer trainable parameters to calibrate pre-trained models on downstream tasks, has been a hot research topic. However, existing PEFT methods within the traditional fine-tuning framework have two main shortcomings: **1)** They overlook the explicit association between trainable parameters and downstream knowledge. **2)** They neglect the interaction between the intrinsic task-agnostic knowledge of pre-trained models and the task-specific knowledge of downstream tasks. These oversights lead to insufficient utilization of knowledge and suboptimal performance. To address these issues, we propose a novel fine-tuning framework, named **GIST**, that can be seamlessly integrated into the current PEFT methods in a plug-and-play manner. Specifically, our framework first introduces a trainable token, called the Gist token, when applying PEFT methods on downstream tasks. This token serves as an aggregator of the task-specific knowledge learned by the PEFT methods and builds an explicit association with downstream tasks. Furthermore, to facilitate explicit interaction between task-agnostic and task-specific knowledge, we introduce the concept of knowledge interaction via a Bidirectional Kullback-Leibler Divergence objective. As a result, PEFT methods within our framework can enable the pre-trained model to understand downstream tasks more comprehensively by fully leveraging both types of knowledge. Extensive experiments on the 35 datasets demonstrate the universality and scalability of our framework. Notably, the PEFT method within our GIST framework achieves up to a 2.25% increase on the VTAB-1K benchmark with an addition of just 0.8K parameters (0.009‰ of ViT-B/16). Code is in the supplementary materials.

## CCS CONCEPTS

• **Computing methodologies** → **Computer vision**; **Natural language processing**.

## KEYWORDS

Parameter efficient fine-tuning, knowledge interaction, vision and language models, few-shot learning

## 1 INTRODUCTION

The advent of large-scale datasets and the pre-training fine-tuning paradigm has empowered pre-trained models to achieve remarkable

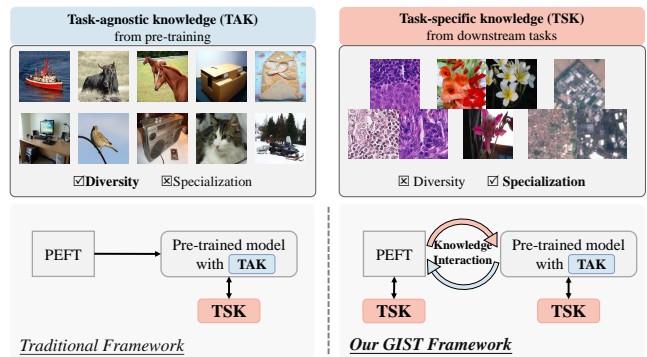

**Figure 1: Unlike the traditional fine-tuning framework, our GIST framework establishes an explicit connection between the learnable PEFT parameters and downstream tasks, thereby comprehensively learning task-specific knowledge (*TSK*). In addition, we introduce the concept of knowledge interaction, establishing interactions between the task-agnostic knowledge (*TAK*) represented by the frozen parameters and the task-specific knowledge (*TSK*) represented by the learnable parameters, thus enabling the model to better adapt to the downstream tasks.**

performances [42]. By leveraging task-agnostic knowledge (**TAK**) from the pre-training phase and learning task-specific knowledge (**TSK**) during the fine-tuning process [31, 47, 50], pre-trained models, particularly Transformer-based models [9, 24], have exhibited exemplary performance across fields such as computer vision (CV) and natural language processing (NLP). However, the burgeoning parameters in Transformer-based models have made the Full parameter fine-Tuning (FT) method less practical for downstream tasks. The FT method necessitates training and retaining separate full parameters for every task. Furthermore, this technique is susceptible to overfitting, especially given the frequently limited data volume in downstream tasks.

With a focus on increasing fine-tuning efficiency, the research community has demonstrated escalating interest in Parameter Efficient Fine-Tuning (PEFT) methods [7]. Briefly, PEFT methods freeze the bulk of pre-trained model parameters, adjusting or introducing a small set of trainable parameters to integrate TSK. However, as shown in Figure 1, PEFT methods within traditional fine-tuning framework do not explicitly establish a connection between the learnable parameters and TSK, and also overlook the interaction with TAK. These deficiencies prevent the model from effectively leveraging these two types of knowledge, thereby leading to suboptimal fine-tuning performance.

To address the above issues, we initiate our investigation from the perspective of TSK acquisition, based on a classic PEFT method (*e.g.*, VPT [15]). Specifically, VPT achieves commendable fine-tuning

performance by freezing the model's backbone parameters and introducing learnable prompt tokens. However, in VPT, prompt tokens are not explicitly used for the final loss calculation, which may impede the full utilization of downstream knowledge. Consequently, we naturally utilize prompt tokens as an additional dependency for computing the loss, obtaining a 0.5% gain. Subsequently, to demonstrate the scalability of this discovery, we attempt to employ this loss as a plug-and-play design. We apply another PEFT method (*e.g.*, Adapter) with VPT, and find that using prompt tokens as an extra criterion also achieves a 0.73% improvement. This discovery implies that naturally using learnable parameters as the basis for computing loss can lead to more effective TSK learning.

Inspired by the above observations, we propose **GIST**, a plug-and-play and efficient framework for existing PEFT methods. Specifically, we first introduce a learnable Gist token ([GIST]). During the fine-tuning phase, [GIST] is co-trained with the learnable PEFT parameters to aggregate the TSK. Akin to the Class token ([CLS]) used to generate Class logits ($S_{cls}$)[1], [GIST] is also processed via the linear classification head, yielding the Gist logits ($S_{gist}$), which subsequently contribute to the loss computation with the true labels. It is imperative to underscore that [CLS] is trained during the pre-training phase to assimilate the TAK, [GIST] undergoes training during the fine-tuning phase, serving as an aggregator for the TSK on downstream tasks.[2] In addition, to engender the interaction between task-agnostic and task-specific knowledge, we employ a Bidirectional Kullback-Leibler Divergence (BKLD) objective, bridging the gap between the two types of knowledge. This bidirectional objective amalgamates both forward and reverse Kullback-Leibler divergence [11] objectives, facilitating a refined computation of the distributional disparity between $S_{cls}$ and $S_{gist}$. In short, via our GIST framework, existing methods could foster better interaction between these two types of knowledge, ensuring the optimal calibration of pre-trained models for enhanced adaptability to downstream tasks.

To sum up, the contributions are as follows: **1)** For the first time, this study analyzes the significance of establishing an explicit connection among learnable parameters, downstream knowledge, and the inherent general knowledge of pre-trained models. This exploration proves to be crucial for understanding the roles of different parameters during the model fine-tuning process. **2)** We introduce the GIST framework, comprising a gist token and a Bidirectional Kullback-Leibler Divergence objective. The former aims to build an association between learnable parameters and downstream tasks, while the latter facilitates deep knowledge interaction. Based on this framework, the model is capable of fully leveraging both general and downstream knowledge to better adapt to downstream tasks. **3)** Extensive experiments are conducted on 35 datasets. The results indicate that the GIST framework can improve fine-tuning performance with minimal increase in the count of trainable parameters.

---

[1]Some vision Transformers, such as Swin Transformer [25], do not introduce an additional [CLS]. Instead, they use the output sequence after global pooling operation for the prediction. In this paper, we uniformly utilize [CLS] for the sake of convenience in expression.

[2]This concept of an aggregator is derived from the pre-training stage of ViT [8], where a [CLS] is introduced to aggregate global information for the final loss calculation. Besides, it is note that [GIST] is introduced during downstream fine-tuning, is trainable, and aggregates TSK, while [CLS] is frozen during fine-tuning and can be considered to retain TAK.

To be specific, when compared to the traditional framework, the PEFT methods implemented within our GIST framework achieve improvements of 1.05%, 1.12%, 1.45%, and 1.1% in image classification, fine-grained few-shot learning, language understanding, and vision&language tasks, respectively.

## 2 RELATED WORKS

### 2.1 Parameter Efficient Fine-tuning

Parameter Efficient Fine-tuning (PEFT) methods enhance the performance of pre-trained models on downstream tasks in a power-saving and efficient manner. Essentially, PEFT techniques modify a select subset or introduce new trainable parameters during fine-tuning to assimilate TSK, thereby calibrating the model's predictions on downstream tasks. Initial explorations into PEFT were predominantly within NLP tasks, with notable methodologies including Adapter [13], Prompt [21], Prefix [22], and LoRA [14], etc. Subsequently, VPT [15] migrates the Prompt technique from NLP to CV, demonstrating the potential of PEFT in visual tasks. For instance, VPT achieves impressive results by fine-tuning with only 0.1% of the total model parameters. AdaptFormer [4] introduces Adapter in parallel into the ViT's FFN layer, achieving performance comparable to the FT method in image recognition and video understanding tasks. SSF [23] adjusts the model's features by scaling and shifting, achieving superior results in image recognition. This success catalyzes further research into PEFT for the CV tasks, with methodologies such as Convpass [16], FacT [17], and ReAdapter [26] advancing the state-of-the-art. However, under the traditional fine-tuning framework, existing PEFT methods do not fully realize their potential because they overlook the explicit connection with TSK and the knowledge interaction with TAK. Therefore, with hardly any increase in parameters, we propose the GIST fine-tuning framework to establish explicit connections and interactions, thereby maximizing the capabilities of existing PEFT methods.

### 2.2 Self-Knowledge Distillation

A concept resonating with our framework is self-knowledge distillation. Knowledge distillation paradigms [12] focus on enhancing the performance of student models by assimilating knowledge from a larger teacher model. In contrast, self-knowledge distillation posits the student model as its own teacher. This is achieved by deriving soft labels through specially crafted branches or distinct distributions, subsequently computing the distillation loss against its own predictions. For instance, in the BYOT approach [48], the deepest classifier is regarded as the teacher, and it imparts its knowledge to shallower networks. CS-KD [45] uses two different samples from the same category to normalize the consistency between two different views of the predicted distribution. USKD [43] utilizes the student model's logits as soft target labels and employs the ranking of intermediate features along with Zipf's law to generate soft non-target labels. Subsequently, USKD performs knowledge distillation using both soft labels from target and non-target classes, making it an advanced approach. In our work, we extrapolate the concept of knowledge distillation. By introducing a learnable token to derive soft labels and employing the BKLD loss as the metric between

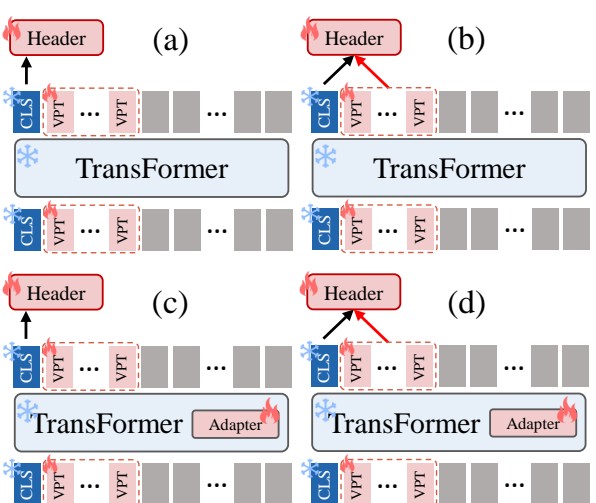

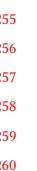

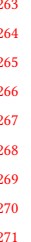

Figure 2: The different fine-tuning structures in Table 1. (a) original VPT in [15]. (b) on the top of (a), additionally using VPT prompt tokens as the basis for calculating loss. (c) combining two classic PEFT methods (VPT and Adapter) for fine-tuning. (d) on the top of (c), additionally using VPT prompt tokens as the basis for calculating loss.

Table 1: Top-1 average accuracy on VTAB-1K. We are progressively experimenting with various structural combinations to enhance performance on downstream tasks. The hidden dimension of the Adapter is 4, and the prompt tokens' length introduced by VPT is 20. It is better viewed in conjunction with Figure 2.

| Tag | Method | Loss | Params. (M) | Mean |
|-----|--------|------|-------------|------|
| (a) | VPT | $\mathcal{L}_{ce}$ | 0.05 | 62.41 |
| (b) | VPT | $\mathcal{L}_{ce} + \mathcal{L}_{vpt}$ | 0.05 | 62.91 |
| - | Adapter | $\mathcal{L}_{ce}$ | 0.13 | 71.46 |
| (c) | Adapter + VPT | $\mathcal{L}_{ce}$ | 0.15 | 71.70 |
| (d) | Adapter + VPT | $\mathcal{L}_{ce} + \mathcal{L}_{vpt}$ | 0.15 | 72.19 |

these soft labels and the model's predictions, our fine-tuning framework aims to augment the efficacy of extant PEFT techniques with negligible parameter overhead.

## 3 METHODS

This section outlines our GIST framework. Initially, in Section 3.1, we reassess the PEFT methods from the perspective of acquiring downstream knowledge, emphasizing the importance of establishing a direct connection between learnable parameters and downstream tasks. Subsequently, Section 3.2 delves into our GIST framework, elucidating how the Gist token is integrated within the Transformer architecture, and how the Bidirectional Kullback-Leibler Divergence (BKLD) loss is employed to facilitate knowledge interaction.

## 3.1 Rethinking PEFT via knowledge acquisition

In this section, we first explore existing PEFT methods from the perspective of knowledge acquisition. Experiments are conducted on the VTAB-1K benchmark, with settings identical to those in Section 4.2. Initially, as shown in Figure 2(a), we start our exploration with a classic PEFT method, VPT-shallow [15]. We fix the length of the learnable prompt tokens introduced by VPT at 20, achieving an accuracy of 62.41% in fine-tuning performance (Table 1(a)). However, in the original VPT, only the Class token is utilized to calculate cross-entropy loss with true labels, and the learnable prompt tokens are not directly involved in the loss computation. We believe this form is suboptimal for fine-tuning phase. Therefore, as shown in Figure 2(b) and Equation 1, we naturally attempt to incorporate the prompt tokens for calculating the cross-entropy loss with the true labels. This simple modification results in a 0.5% increase in fine-tuning performance (Table 1(b)). A possible reason is that explicitly including learnable parameters in the loss calculation can lead to more comprehensive task-specific knowledge (**TSK**) acquisition.

$$\mathcal{L} = \mathcal{L}_{ce}(S_{cls}, y) + \mathcal{L}_{vpt}$$
$$\mathcal{L}_{vpt} = \mathcal{L}_{ce}(S_{vpt}, y)$$
(1)

where $\mathcal{L}_{ce}$ denotes the cross-entropy loss, and $y$ represents true labels. $S_{cls}$ and $S_{vpt}$ represent the logits obtained from the Class token and VPT prompt tokens after passing through the linear classification head, respectively.

Subsequently, we investigate the feasibility of integrating the loss calculation approach derived from VPT with other PEFT methods. As shown in Figures 2(c, d), VPT is implemented alongside the Adapter for the fine-tuning process. Performance is evaluated and compared before and after the incorporation of the additional loss $\mathcal{L}_{vpt}$. The results indicate that this supplementary loss, detailed in Tables 1(c, d), further enhances the Adapter's fine-tuning efficacy.

Therefore, we pose a question: *Can this method serve as a free lunch-style framework to enhance the fine-tuning performance of existing PEFT methods?* The answer is affirmative. In the next section, we introduce our GIST fine-tuning framework, which can improve the performance of PEFT methods in a plug-and-play manner without adding extra burden.

## 3.2 GIST Framework

As discussed in Section 3.1, incorporating VPT's prompt tokens for additional loss calculation can enhance fine-tuning performance. However, this approach also introduces an increased parameter burden. Additionally, relying solely on $\mathcal{L}_{vpt}$ as an extra loss component does not fully utilize the task-agnostic knowledge (**TAK**) from the pre-training phase. Therefore, as shown in Figure 3, our GIST framework introduces a learnable token called the Gist token, which is only 1 in length and designed to be an aggregator for TSK. Furthermore, we introduce the BKLD loss for knowledge interaction, thereby maximizing the potential of PEFT methods for more effective fine-tuning.

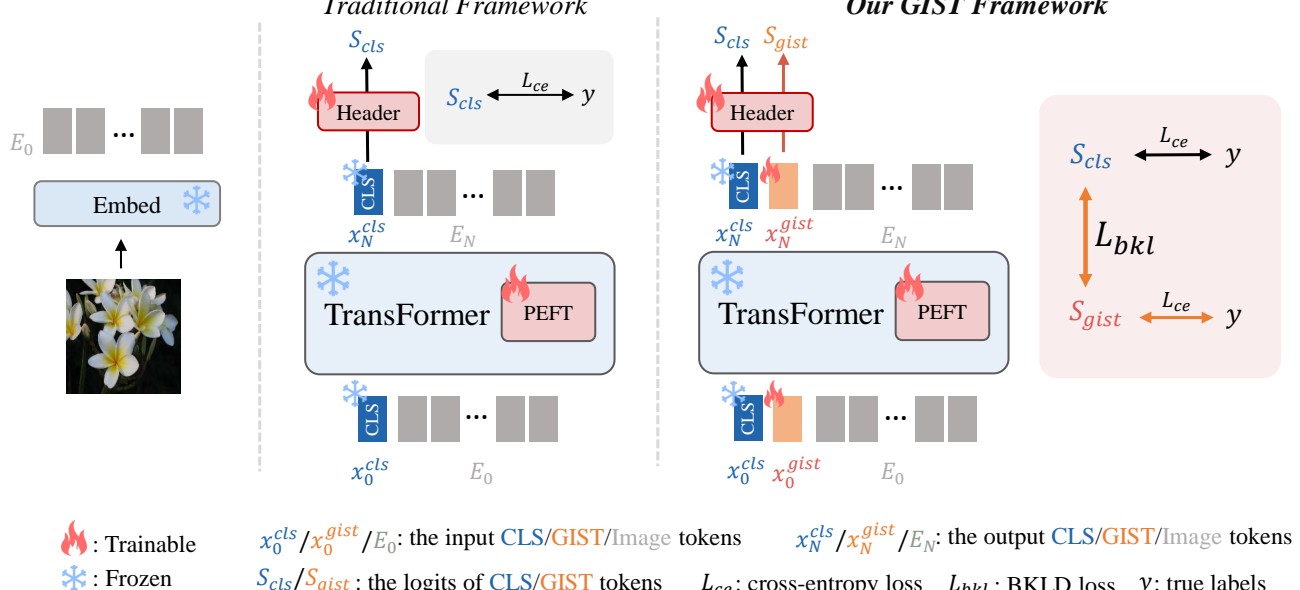

**Traditional Framework** · **Our GIST Framework**

🔥 : Trainable    $x_0^{cls}/x_0^{gist}/E_0$: the input CLS/GIST/Image tokens    $x_N^{cls}/x_N^{gist}/E_N$: the output CLS/GIST/Image tokens

❄ : Frozen    $S_{cls}/S_{gist}$: the logits of CLS/GIST tokens    $L_{ce}$: cross-entropy loss    $L_{bkl}$: BKLD loss    $y$: true labels

**Figure 3: Overview of our GIST fine-tuning framework. Unlike the traditional fine-tuning framework, we introduce a learnable "Gist" token, which collaboratively learns task-specific knowledge with the PEFT method on downstream tasks. Subsequently, we introduce a Bidirectional Kullback-Leibler Divergence loss to facilitate knowledge interaction.**

*3.2.1 Gist token ([GIST]).* For a Transformer model[3], the input data undergoes an embedding transformation, yielding a sequence $x \in \mathbb{R}^{L \times D}$. Afterwards, the Class token $x_0^{cls} \in \mathbb{R}^{1 \times D}$ is concatenated with this sequence, augmented by a positional embedding $P \in \mathbb{R}^{(L+1) \times D}$, resulting in the input sequence $X_0 \in \mathbb{R}^{(L+1) \times D}$, as formulated in Equation 2.

$$X_0 = [x_0^{cls}; x] + P \tag{2}$$

where $[\cdot; \cdot]$ represents the concatenation operation. Subsequent processing of $X_0$ ensues through a series of Transformer layers, as depicted in Equation 3.

$$\begin{aligned} X_l' &= \text{MHSA}(\text{LN}(X_{l-1})) + X_{l-1} \\ X_l &= \text{FFN}(\text{LN}(X_l')) + X_l', l = 1, 2, ..., N \end{aligned} \tag{3}$$

where MHSA stands for multi-head self-attention block, FFN represents the feed-forward network, and LN stands for LayerNorm [2]. After applying all Transformer layers, we can derive $x_N^{cls}$ from $X_N$. The logits $S_{cls}$ can then be obtained using the trainable linear classification head (HEAD), as shown in Equation 4.

$$S_{cls} = \text{HEAD}(x_N^{cls}) \tag{4}$$

Ultimately, the cross entropy loss function computes the loss between $S_{cls}$ and the true labels. Notably, during the fine-tuning phase, $x_0^{cls}$ is frozen, preserving the model's TAK from the pre-training phase. Different from the traditional fine-tuning framework as shown in Figure 3, our framework introduces an additional

[3]Our GIST framework does not alter the implementation of the existing PEFT methods. Therefore, for the sake of simplicity in expression, we omit the processing procedures of the PEFT methods in this section.

learnable token ([GIST]), denoted as $x_0^{gist} \in \mathbb{R}^{1 \times D}$, to aggregate the TSK learned by the PEFT method during fine-tuning. Thus, on the basis of Equation 2, we concatenate [GIST] to obtain our input sequence $X_0 \in \mathbb{R}^{(L+2) \times D}$, as Equation 5.

$$X_0 = [[x_0^{cls}; x] + P; x_0^{gist}] \tag{5}$$

After processing the input sequence through all Transformer layers, $x_N^{cls}$ and $x_N^{gist}$ are derived from $X_N$. We subsequently send both $x_N^{cls}$ and $x_N^{gist}$ through the trainable linear classification head, resulting in $S_{cls}$ and $S_{gist}$, respectively. The loss $\mathcal{L}_{cls}$ is computed by contrasting $S_{cls}$ with the truth labels. Similarly, the loss $\mathcal{L}_{gist}$ is determined by comparing $S_{gist}$ with the true labels. These two loss terms can be expressed as follows:

$$\begin{aligned} \mathcal{L}_{cls} &= \mathcal{L}_{ce}(S_{cls}, y) \\ \mathcal{L}_{gist} &= \mathcal{L}_{ce}(S_{gist}, y) \end{aligned} \tag{6}$$

where $\mathcal{L}_{ce}$ is the cross entropy loss, $y$ is the true labels.

*3.2.2 Bidirectional Kullback-Leibler Divergence (BKLD) Loss.* Only utilizing $\mathcal{L}_{cls}$ and $\mathcal{L}_{gist}$ does not facilitate explicit interaction between the TAK represented by $S_{cls}$ and the TSK represented by $S_{gist}$. Therefore, we introduce the BKLD loss function, as shown in Equation 7.

$$\begin{aligned} \mathcal{L}_{bkl} &= \mathcal{L}_{fkl} + \mathcal{L}_{rkl} \\ &= \text{KL}(S_{cls} \| S_{gist}; T) + \text{KL}(S_{gist} \| S_{cls}; T) \end{aligned} \tag{7}$$

where $\mathcal{L}_{bkl}$ represents our BKLD loss. $\mathcal{L}_{fkl}$ is the forward KLD loss. $\mathcal{L}_{rkl}$ is the reverse KLD loss. $\text{KL}(\cdot \| \cdot; T)$ means computing the KL divergence between two distributions with a temperature $T$. The parameter $T$, is introduced to soften the outputs before

they are processed through softmax, adjusting the sharpness of the distribution. Higher values of $T$ produce softer probabilities [10].

For most knowledge distillation methods, the forward KLD is generally utilized as the loss function. It can be represented as $\mathcal{L}_{fkl} = \text{KL}(p||q;T)$, where $p$ and $q$ represent two different distributions. With $p$ taken as the reference, $\mathcal{L}_{fkl}$ quantifies how much the distribution $q$ diverges from $p$. Conversely, the reverse KLD, denoted as $\mathcal{L}_{rkl} = \text{KL}(q||p;T)$, uses $q$ as the reference and measures the divergence of distribution $p$ from $q$. In this paper, we leverage both forward and reverse KLD as loss functions to facilitate explicit interaction between TAK and TSK. On one hand, we employ the forward KLD loss to enhance the learning of TSK, guided by TAK. On the other hand, by utilizing the reverse KLD loss, we ensure that the pre-trained model is more effectively tailored to downstream tasks, following the directives of TSK.

*3.2.3 Overall Loss.* The overall loss function is derived by amalgamating $\mathcal{L}_{cls}$, $\mathcal{L}_{gist}$, and $\mathcal{L}_{bkl}$, as depicted in Equation 8. This loss function guides the model during the fine-tuning phase, allowing [GIST] to co-learn with the PEFT parameters and aggregate TSK, while fully leveraging TAK to ensure an explicit interaction between the two types of knowledge.

$$\mathcal{L}_{all} = \mathcal{L}_{cls} + \mu\mathcal{L}_{gist} + \lambda\mathcal{L}_{bkl} \qquad (8)$$

where $\mu$ and $\lambda$ is the hyperparameter that controls the trade-off among the three loss terms. It is noteworthy that the aforementioned use of $S_{gist}$ is limited only to the training process. For the inference process, we still solely rely on $S_{cls}$ as the exclusive basis for prediction.

# 4 EXPERIMENTS

## 4.1 Datasets and metrics

*4.1.1 Image classification tasks.* We utilize the VTAB-1K benchmark [46] to validate our GIST framework for image classification tasks. Specifically, VTAB-1K includes 19 different datasets, which can be categorized into three groups: Natural, Specialized, and Structured. Each dataset consists of 1,000 samples for training, with an average of 20,000 samples for testing, making it a highly challenging benchmark. Following previous works [23], for each dataset, we report the Top-1 accuracy on the test set. For the entire benchmark, we present the arithmetic mean of the Top-1 accuracy.

*4.1.2 Fine-grained few-shot tasks.* In a few-shot setting, we validate the performance of our framework in the low-data regime using Food-101 [3], OxfordPets [32], Stanford Cars [20], Oxford-Flowers102 [29], and FGVC-Aircraft [27] datasets. Similar to previous work [17, 49], we conduct validation under {1, 2, 4, 8, 16}-shot settings and report the Top-1 accuracy on the test set.

*4.1.3 Language understanding tasks.* To validate the universality of our framework, we also conduct verification for the PEFT method in NLP. GLUE benchmark [38] is utilized to verify the effectiveness of GIST framework. Specifically, we train and test on a total of 8 tasks: MNLI, QQP, QNLI, SST-2, STS-B, MRPC, RTE, and CoLA. Following previous works [1], we use Pearson Correlation for STS-B and accuracy for other tasks as metrics.

*4.1.4 Vision&Language tasks.* : To further illustrate the universality of our fine-tuning framework, we validate the PEFT method in the multi-modal and few-shot scenarios. We conduct few-shot training on the Flowers102 [30], DTD [5], and UCF101 [36] datasets, which involve image classification tasks accompanied by corresponding text. Following previous research [19], we employ three different random seeds and report the mean Top-1 accuracy. In addition, for a more detailed introduction, please refer to the Appendix in the supplementary materials.

## 4.2 Implementation details

For the **VTAB-1K** benchmark and **FGVC** datasets, we employ the ViT-B/16 [8] model, pre-trained on the ImageNet-21K dataset [6], as the backbone. In terms of training configurations, we follow the work of predecessors [17, 18, 23, 26], to ensure fairness and reproducibility. Turning to the **GLUE** benchmark, we harness the T5-base [35] model as the backbone. Similar to the setting of the previous work [1] by configuring a batch size of 32, imposing a maximum token length of 256, setting the learning rate to 3e-4, and conducting training for 20 epochs on each task. In the case of **Vision&Language tasks** (Flowers102, DTD, and UCF101 datasets), we utilize a 16-shot setup training regimen, subsequently evaluating performance on the full test sets. Consistent with the prior research [19], we maintain the same training settings.

Regarding our GIST framework, to avoid redundancy brought about by further hyperparameter adjustment, we fix temperature $T$ at 3, $\mu$ to 0.5, and only allow $\lambda$ to be searched from {0.25, 0.5, 0.75}. Pytorch [33] and Transformers [40] packages are utilized to implement experiments on NVIDIA RTX 3090 GPUs and NVIDIA A100 GPUs, and more detailed settings are in the Appendix.

## 4.3 Main results

*4.3.1 Comparative Results on VTAB-1K.* We have thoroughly validated GIST framework on the benchmark for visual tasks, and the experimental results are shown in Table 2. For the three types of PEFT methods summarized by [44], namely Adapter Tuning (Adapter, ReAdapter [26] and Bi-Adapter [18]), Prompt Tuning (VPT [15]), and Parameter Tuning (LoRA [14], SSF [23] and FacT [17]), we apply these methods within our framework for fine-tuning on downstream tasks. This further improves the performance of the existing PEFT methods with an average increase of 1.05%, without significantly increasing the number of parameters. In the best case, our GIST can improve Adapter's performance by 2.25%, and in the worst case, it can still enhance FacT's performance by 0.32%. The results indicate that our framework facilitates a more comprehensive knowledge interaction and enhances the performance of PEFT methods by fully leveraging the two types of knowledge.

*4.3.2 Comparative Results on FGVC.* We conduct thorough validation in a few-shot scenario for fine-grained recognition. The PEFT methods used are Adapter, VPT, and SSF, which are fine-tuned under both the traditional framework and our GIST framework, with results shown in Figure 4. Overall, even in the low-regime few-shot scenario, fine-tuning different types of PEFT methods under the GIST framework can improve performance without significantly increasing the number of trainable parameters.

Table 2: The comparative results on the VTAB-1K benchmark. The symbol * indicates employing the PEFT method within our GIST framework. FT represents the full parameter fine-tuning method, and LP stands for the Linear Probing method. Params. stands for trainable parameters.

| | Natural | | | | | | | Specialized | | | | Structured | | | | | | | | | | |
| Method | CIFAR-100 | Caltech101 | DTD | Flowers102 | Pets | SVHN | Sun397 | Patch Camelyon | EuroSAT | Resisc45 | Retinopathy | Clevr/count | Clevr/distance | DMLab | KITTI/distance | dSprites/loc | dSprites/ori | SmallNORB/azi | SmallNORB/ele | Mean (%) | Δ | Params. (M) |
|---|---|---|---|---|---|---|---|---|---|---|---|---|---|---|---|---|---|---|---|---|---|---|
| FT | 68.9 | 87.7 | 64.3 | 97.2 | 86.9 | 87.4 | 38.8 | 79.7 | 95.7 | 84.2 | 73.9 | 56.3 | 58.6 | 41.7 | 65.5 | 57.5 | 46.7 | 25.7 | 29.1 | 65.57 | - | 85.84 |
| LP | 63.4 | 85.0 | 63.2 | 97.0 | 86.3 | 36.6 | 51.0 | 78.5 | 87.5 | 68.6 | 74.0 | 34.3 | 30.6 | 33.2 | 55.4 | 12.5 | 20.0 | 9.6 | 19.2 | 52.94 | - | 0.04 |
| Adapter | 70.2 | 92.6 | 74.6 | 99.4 | 91.2 | 80.4 | 51.4 | 84.1 | 96.3 | **88.0** | 75.6 | **84.2** | 59.6 | 53.2 | 76.3 | 60.7 | 51.9 | 27.8 | 40.2 | 71.46 | 2.25↑ | 0.13 |
| **Adapter*** | 74.5 | 92.3 | 76.9 | 99.5 | 92.3 | 85.7 | 54.6 | 88.2 | 96.5 | 87.9 | **77.4** | 83.6 | 61.2 | 54.0 | 81.2 | 72.3 | 52.1 | 29.3 | 41.0 | 73.71 | 2.25↑ | 0.13 |
| ReAdapter | 72.4 | 91.6 | 71.0 | 99.2 | 91.4 | 90.7 | 55.1 | 85.3 | 95.9 | 84.6 | 75.9 | 82.3 | 68.0 | 50.4 | 79.9 | 80.4 | 49.2 | **38.6** | 41.0 | 73.83 | 0.43↑ | 0.22 |
| **ReAdapter*** | 73.4 | 92.7 | 71.5 | 99.2 | 91.5 | **91.4** | 55.4 | 84.9 | 96.3 | 85.2 | 75.6 | 82.6 | **70.2** | 51.2 | 80.9 | 82.0 | 47.3 | 36.9 | 42.8 | 74.26 | 0.43↑ | 0.22 |
| Bi-Adapter | 74.1 | 92.4 | 72.1 | 99.3 | 91.6 | 89 | 56.3 | **88.2** | 95.2 | 86.0 | 76.2 | 83.9 | 63.6 | 53.0 | **81.4** | 86.2 | 54.8 | 35.2 | 41.3 | 74.76 | 0.42↑ | 0.11 |
| **Bi-Adapter*** | 74.5 | **94.4** | 72.5 | 99.4 | 91.7 | 89.6 | **56.9** | 88.1 | 95.6 | 86.5 | 76.0 | 84.0 | 63.1 | 53.0 | **81.4** | **87.8** | 53.1 | 36.5 | **43.7** | **75.18** | 0.42↑ | 0.11 |
| VPT | 60.5 | 90.6 | 70.6 | 99.1 | 89.3 | 50.1 | 50.8 | 82.2 | 93.8 | 82.5 | 74.9 | 50.6 | 58.9 | 41.0 | 68.1 | 39.0 | 32.4 | 22.3 | 29.1 | 62.41 | 1.22↑ | 0.05 |
| **VPT*** | 64.6 | 90.9 | 72.3 | 99.3 | 90.4 | 56.4 | 52.6 | 82.8 | 93.9 | 83.6 | 75.1 | 49.0 | 60.5 | 41.1 | 66.9 | 43.0 | 34.8 | 22.7 | 29.1 | 63.63 | 1.22↑ | 0.05 |
| LoRA | 65.9 | 91.3 | 73.6 | 99.3 | 91.7 | 83.2 | 51.0 | 84.0 | 96.2 | 87.3 | 76.2 | 71.2 | 57.8 | 50.5 | 78.1 | 58.4 | 53.2 | 28.1 | 41.1 | 70.43 | 1.78↑ | 0.06 |
| **LoRA*** | 72.0 | 91.4 | 76.6 | 99.6 | 91.9 | 85.2 | 55.1 | 86.0 | 96.1 | 87.1 | 76.8 | 76.0 | 59.3 | 50.8 | 78.7 | 63.2 | 53.9 | 29.3 | 42.9 | 72.21 | 1.78↑ | 0.06 |
| SSF | 69.0 | 92.6 | 75.1 | 99.4 | 91.8 | 90.2 | 52.9 | 87.4 | 95.9 | 87.4 | 75.5 | 75.9 | 62.3 | 53.3 | 80.6 | 77.3 | **54.9** | 29.5 | 37.9 | 73.10 | 0.91↑ | 0.24 |
| **SSF*** | 74.2 | 93.1 | 74.4 | **99.5** | 91.8 | 91.2 | 53.7 | 87.5 | 96.1 | 87.3 | 76.2 | 79.1 | 61.6 | **54.5** | 81.2 | 81.7 | 53.9 | 30.9 | 38.2 | 74.01 | 0.91↑ | 0.24 |
| FacT | 70.6 | 90.6 | 70.8 | 99.1 | 90.7 | 88.6 | 54.1 | 84.8 | 96.2 | 84.5 | 75.7 | 82.6 | 68.2 | 49.8 | 80.7 | 80.8 | 47.4 | 33.2 | 43.0 | 73.23 | 0.32↑ | 0.11 |
| **FacT*** | 71.0 | 91.8 | 70.2 | 99.0 | 90.8 | 89.3 | 54.1 | 85.7 | 95.5 | 84.3 | 75.6 | 83.2 | 69.2 | 50.3 | 80.2 | 81.4 | 47.6 | 35.2 | 43.1 | 73.55 | 0.32↑ | 0.11 |

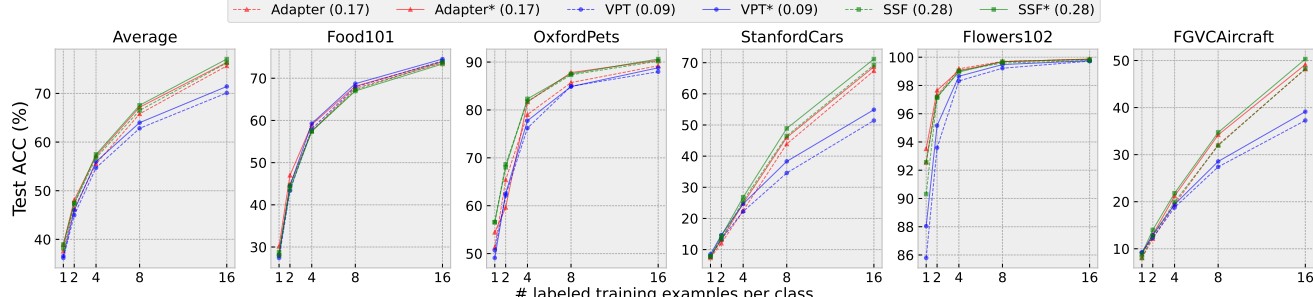

Figure 4: Top-1 accuracy of few-shot learning on FGVC datasets. The trainable parameters (M) is shown in parentheses.

*4.3.3 Comparative Results on GLUE.* We conduct validation on the GLUE benchmark for NLP tasks, and the results are shown in Table 3. When applying the FT method, 220M parameters are used to achieve a performance of 84.95%. When applying Adapter [13] within the traditional fine-tuning framework, 1.9M parameters are needed, but the performance is still 0.5% lower than that of the FT method. Notably, when utilizing Adapter within our fine-tuning framework, the performance improves by 1.45%, even exceeding the FT method by 0.95%.

*4.3.4 Comparative Results on Vision&Language tasks.* We further conduct experiments for PEFT method in the multi-modal field, training with a 16-shot setup, and compare the results with zero-shot CLIP [34] and MaPLe [19], as shown in Table 4. When applied

to few-shot scenarios in a multi-modal context, GIST also demonstrates promising performance. Incorporating MaPLe into our fine-tuning framework yields a 1.10% enhancement compared to the original MaPLe, with almost no additional parameters introduced. Besides, this performance significantly surpasses that of CLIP.

## 4.4 Ablation studies

We conduct extensive ablation experiments on the VTAB-1K benchmark. Unless otherwise specified, we employ the ViT-B/16 model, pre-trained on the ImageNet-21K dataset, as the backbone, and use Adapter as the PEFT method. Furthermore, the symbol * indicates employing the PEFT method within our GIST framework, and we

**Table 3: The comparative results on the GLUE benckmark. We use Pearson Correlation for STS-B, and accuracy for other tasks as metrics. The symbol ∗ indicates employing the PEFT method within our GIST framework.**

| Method | MNLI | QQP | QNLI | SST-2 | STS-B | MRPC | RTE | CoLA | Mean | Params. (M) |
|---|---|---|---|---|---|---|---|---|---|---|
| FT | 86.8 | **91.6** | 93.0 | **94.6** | 89.7 | **90.2** | 71.9 | 61.8 | 84.95 | 220 |
| Adapter | 86.5 | 90.2 | **93.2** | 93.8 | 90.7 | 85.3 | 71.9 | 64.0 | 84.45 | 1.9 |
| **Adapter∗** | **86.9** | 90.6 | **93.2** | 94.0 | **90.8** | 88.7 | **77.7** | **65.3** | **85.90** | 1.9 |

**Table 4: The comparative results on the Vision&Language tasks under the 16-shot setting.**

| Method | Flowers102 | DTD | UCF101 | Mean | Params. (M) |
|---|---|---|---|---|---|
| CLIP | 71.30 | 44.56 | 66.72 | 60.86 | - |
| MaPLe | 94.03 | 67.79 | 81.18 | 81.00 | 3.56 |
| **MaPLe∗** | **95.04** | **69.23** | **82.02** | **82.10** | 3.56 |

**Table 5: Ablation studies for different $\lambda$.**

| $\lambda$ | Mean |
|---|---|
| - | 71.46 |
| 0.25 | 73.31 |
| 0.5 | 73.18 |
| 0.75 | **73.44** |

**Table 6: Ablation studies for different Gist token length.**

| token len. | Mean | Params. (M) |
|---|---|---|
| 1 | **73.71** | 0.13 |
| 10 | 73.42 | 0.14 |
| 50 | 71.96 | 0.16 |
| 100 | 71.21 | 0.21 |

**Table 7: Ablation results on our loss function.**

| $\mathcal{L}_{cls}$ | $\mathcal{L}_{gist}$ | $\mathcal{L}_{bkl}$ | Mean |
|---|---|---|---|
| ✓ | | | 71.46 |
| ✓ | ✓ | | 72.71 |
| ✓ | | ✓ | 73.29 |
| ✓ | ✓ | ✓ | **73.71** |

**Table 8: Results on different loss functions for knowledge interaction.**

| Loss function | Mean |
|---|---|
| $\mathcal{L}_{cls}$ | 71.46 |
| $\mathcal{L}_{cls}+\mathcal{L}_{gist}+\mathcal{L}_{mse}$ | 73.03 |
| $\mathcal{L}_{cls}+\mathcal{L}_{gist}+\mathcal{L}_{cos}$ | 72.88 |
| $\mathcal{L}_{cls}+\mathcal{L}_{gist}+\mathcal{L}_{bkl}$ | **73.71** |

display the arithmetic mean of the Top-1 accuracy. More detailed results can be found in the Appendix.

*4.4.1 The impact of $\lambda$.* In our GIST framework, we only search for $\lambda$ from the set $\{0.25, 0.5, 0.75\}$ to control the interaction strength between task-agnostic and task-specific knowledge. Therefore, we first conduct ablation experiments for different interaction strengths, and the results are shown in Table 5. The results indicate that regardless of the interaction strength, our fine-tuning framework can further enhance the performance. Even in the worst case with $\lambda = 0.5$, there's still an improvement of nearly 2%.

*4.4.2 The impact of token length.* As illustrated in Table 6, we evaluate the performance of the GIST framework with varying lengths of the Gist token. The results indicate a clear trend: as the token length increases, the effectiveness of our fine-tuning framework decreases.

**Table 9: Results on ViT-S/16 (S) and ViT-L/16 (L).**

| Method | Params | Mean |
|---|---|---|
| S+Adapter | 0.07 | 71.39 |
| **S+Adapter∗** | 0.07 | 72.47 |
| L+Adapter | 0.30 | 71.81 |
| **L+Adapter∗** | 0.30 | **73.89** |

**Table 10: Ablation results on Swin-B.**

| Method | Params | Mean |
|---|---|---|
| FT | 86.7 | 72.46 |
| Linear probing | 0.1 | 58.19 |
| Adapter | 0.21 | 73.19 |
| **Adapter∗** | 0.21 | **74.15** |

This is analogous to the role of the Class token during pre-training, which accumulates task-agnostic knowledge from diverse training data. Similarly, the Gist token is designed to aggregate task-specific knowledge during the fine-tuning phase. It is important to note that the Class token's length is fixed at one, whereas increasing the Gist token's length may lead to disproportionate knowledge interaction and a subsequent decline in performance.

*4.4.3 The impact of loss function.* In this study, we employ loss functions that extend beyond traditional classification loss, encompassing two components: $\mathcal{L}_{gist}$ and $\mathcal{L}_{bkl}$. To evaluate the individual contributions of these components, we execute ablation studies, the results are presented in Table 7. Evidently, the efficacy of the GIST framework diminishes with a reduction in the number of loss terms. We first demonstrate the importance of establishing a direct connection between learnable parameters and task-specific knowledge during the fine-tuning process. When we introduce $\mathcal{L}_{gist}$ into the basic loss function $\mathcal{L}_{cls}$, the accuracy improved by 1.25%. Alternatively, by adding $\mathcal{L}_{bkl}$ to $\mathcal{L}_{cls}$, it achieves a performance gain of 1.83%, underscoring the effectiveness of knowledge interaction during downstream fine-tuning. Finally, when we introduce both types of losses simultaneously, the overall performance improves by 2.25%. This not only proves the compatibility of these two loss functions but also indicates that more comprehensive downstream knowledge acquisition can enhance the effects of knowledge interaction.

Furthermore, we assess the performance of our framework by substituting the BKLD loss with the Mean Squared Error loss $\mathcal{L}_{mse}$ and the Cosine Similarity loss $\mathcal{L}_{cos}$. The comparative results are depicted in Table 8. Intriguingly, within the confines of our GIST framework, replacing our BKLD loss by common loss functions for knowledge interaction still yields a performance enhancement ranging from 1% to 2%. This attests to the scalability of our fine-tuning framework. Namely, when more advanced loss functions are proposed in subsequent research, our GIST framework can also be utilized directly to enhance the performance of the PEFT methods.

*4.4.4 The impact of different networks.* First, to illustrate the versatility of our GIST across models of varying sizes, we substitute ViT-B/16 with ViT-S/16 and ViT-L/16, as detailed in Table 9. Next, to highlight our framework's adaptability to different network structures, we conduct experiments using Swin-B [25] as the backbone, as presented in Table 10. As evident from Tables 9 and 10, regardless of whether we modify the model size or transition to an alternate backbone, our GIST consistently bolsters performance without a significant increase in parameters.

**Table 11: The comparative results with different self-knowledge distillation methods.**

| Method | Mean | Natural | Specialized | Structured |
|---|---|---|---|---|
| Adapter | 71.46 | 79.96 | 86.02 | 56.73 |
| Adapter+BYOT | 69.70 | 77.86 | 86.24 | 54.29 |
| Adapter+CS-KD | 71.24 | **82.63** | 86.22 | 53.78 |
| Adapter+USKD | 71.40 | 80.14 | 86.78 | 56.06 |
| **Adapter*** | **73.71** | 82.26 | **87.50** | **59.24** |

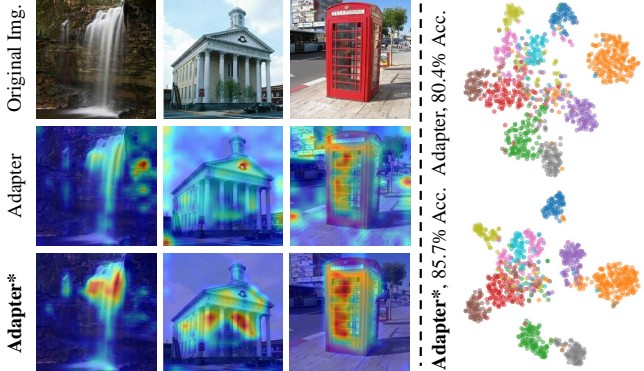

**Figure 5:** *Left:* **The attention map visualization on Sun397 dataset.** *Right:* **The t-SNE visualization on SVHN dataset.**

*4.4.5 Comparisons with self-knowledge distillation methods.* In this study, we compare our approach with two classical methods (BYOT [48] and CS-KD [45]) as well as a state-of-the-art method (USKD [43]). The results are presented in Table 11, which reveal that even the most advanced self-knowledge distillation (SKD) methods can lead to a performance degradation of PEFT methods. A potential reason is that the existing SKD methods do not specifically acquire soft labels tailored for fine-tuning phase. In contrast to them, the Gist token we introduced serves as an aggregator, effectively capturing task-specific knowledge, thereby providing superior soft labels for knowledge interaction.

### 4.5 Visualization

We conduct attention map and t-SNE [37] visualization analysis, as depicted in Figure 5. For this, we extract the [CLS] following the final Transformer layer and preceding the linear classification head. This analysis is performed on the Sun397 [41] and SVHN [28] dataset. Notably, upon integrating GIST, the attention is more focused on the target object, and the classification clusters appear more condensed. This suggests that our framework enhances the ability of existing PEFT methods to assimilate more thorough task-specific knowledge via knowledge interaction. More results of visualization are in the Appendix.

### 4.6 Analysis

In this section, we explore the underlying factors contributing to the performance enhancements offered by the GIST framework. As illustrated in Figure 6, using the Retinopathy dataset, we depict the

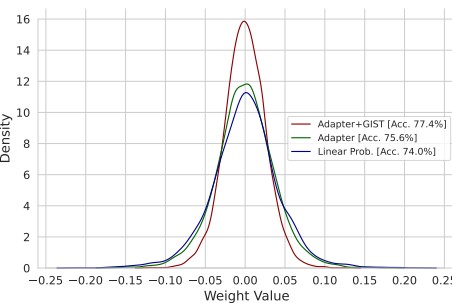

**Figure 6: The weight distribution visualization on Retinopathy dataset.**

distribution of the weights from the linear classification head after fine-tuning. In general, a narrower distribution of weights is commonly associated with enhanced robustness. Minor adjustments to weights suggest that the model is less prone to overly relying on noise or irrelevant features from the inputs, thus diminishing the risk of overfitting [39]. The GIST framework results in an even more compact weight distribution, potentially serving as a form of regularization. Furthermore, the efficacy of combining PEFT methods with regularization effects warrants further exploration in future research.

### 4.7 Discussion

Our GIST framework possesses the following two preeminent characteristics: **1) Universality**: In the experimental section, we conduct experiments for PEFT methods on the image classification, fine-grained few-shot, language understanding and vision&language tasks. The results demonstrate that our framework is versatile and can be applied to PEFT methods across various scenarios, not just confined to computer vision fields. **2) Scalability**: At the core of GIST framework lies the principle of knowledge interaction, which can be realized in multiple ways, not merely limited to the approach presented in this paper. A simple illustration, as shown in Table 8, reveals that by substituting the BKLD loss with other common losses for knowledge interaction, performance can still be augmented. This means that advanced loss functions in future research can be seamlessly integrated into our GIST framework.

## 5 CONCLUSIONS

In this paper, we propose GIST, an efficient and straightforward fine-tuning framework, tailored specifically for PEFT methods. This framework incorporates a learnable Gist token to explicitly establish a connection between trainable parameters and downstream tasks, thereby aiming to acquire a more comprehensive task-specific knowledge. In addition, it employs a Bidirectional Kullback-Leibler Divergence loss to enhance the interaction between task-specific and intrinsic task-agnostic knowledge of pre-trained models. Extensive experiments demonstrate that integrating existing PEFT methods with our GIST framework leads to improved performance without significantly increasing the parameter count. Notably, the framework's universality and scalability make it exceptionally suitable for a broad range of scenarios.

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
