# OpenReview forum: "GIST: Improving Parameter Efficient Fine-Tuning via Knowledge Interaction"
_acmmm.org/ACMMM/2024/Conference — MM2024 Poster_

### Official Review · Reviewer_CzCb · 2024-05-06

**Rating:** 3
**Confidence:** 3

**Summary:**

This work proposes a novel fine-tuning framewort to seamlessly integrate into the current PEFT methods in a plug-and-play manner, which analyses task-specific knowledge learned by the PEFT methods and builds an explicit association with downstream tasks. Extensive experiments on the various datasets demonstrate the universality and scalability of their framework.

**Strengths:**

1. The method relies on self-distillation to learn task knowledge, is interesting and meaningful for the PEFT.

2. The authors report performance in much widely used datasets, and their comparisons cover the latest works in PEFT. These comparisons show that the proposed method  can improve fine-tuning performance.

**Limitations:**

1. In existing works, the most similar one to this manuscript is VPT[1], which introduces the fine-tune method to adapt large pre-trained Transformers for downstream tasks. Meanwhile, for exploiting the task-agnostic and  task-specific knowledge, many self-distilling methods(e.g.,[2].[3] ) also use the task-related knowledge to improve model performance for various tasks. This paper combines two kind of approach. Both works already exist, and the approach section in the proposed paper is just a reiteration of the explanation and equations in the previous work.

2.Ablation study on different downstream data scales is missed. Also what about the time cost compared with otther PEFT method?

3.The experiment is inadequate. The author only compare other methods based on ViT-B model, but what about other scales model[1](e.g., ViT-L and ViT-H)?

Adding/solving the main concerns can increase this paper's chances for publication.

[1]Visual Prompt Tuning.(ECCV22)

[2]From Knowledge Distillation to Self-Knowledge Distillation: A Unified Approach with Normalized Loss and Customized Soft Labels(ICCV23)

[3]Regularizing Class-wise Predictions via Self-knowledge Distillation（CVPR20）

**Suitability:**

2

---

### Official Review · Reviewer_pmtt · 2024-05-17

**Rating:** 4
**Confidence:** 3

**Summary:**

This paper proposed a plug-and-play Parameter Efficient Fine-Tuning technique by introducing a learnable token and by including this token in the calculation of the loss function. Specifically, (1) the learnable [GIST] token acquired Task-Specific Knowledge by calculating a cross-entropy function with the true labels during fine-tuning; (2) Task-Agnostic Knowledge acquired during pre-training is transferred to the fine-tuning via a knowledge interaction mechanism, which is implemented by decreasing the distance between CLS logits and GIST logits. Extensive experiments have shown the universality and scalability of the proposed method. And it’s noticeable that the proposed method has enhanced fine-tuning performances with negligible parameter overhead.

**Strengths:**

(1) This article is clearly written, easy to understand, and logically coherent.

(2) The proposed method is in a plug-and-play manner, which could be leveraged as a general regularization method for various framework.

(3) The experiments are comprehensive and thorough.

(4) The motivation/insight, which is explained by including prompt tokens in loss calculation, is helpful since it carefully underlined an important detail in the fine-tuning. And the proposed method is helpful for understanding the roles of different parameters during the model fine-tuning process.

**Limitations:**

(1) The reason behind the improvement discussed in 4.7 is not convincing and seems too simple. A more comprehensive or theoretical analysis is expected for such a simple method.

(2) Discussion for some negative experimental results is neglected. Some decrease in performance can be observed, e.g., in Table 2, Adapter method on Clevr/count benchmark and Bi-Adapter method on Clevr/distance benchmark; in Figure 4, SSF method on Food101. The decrease means that we introduced more parameters but result in a worse performance, so it’s really negative and worth discussing.

(3) Averaging along the "Method" dimension actually diminishes the reader's perception of certain negative results (although acceptable). Therefore, objectively, I recommend that the authors provide another table in the appendix, indicating which methods/datasets show improvements or deteriorations in performance. Additionally, statistical significance should be demonstrated to evaluate the value of the proposed method (considering that extra parameters are still introduced in this paper, if the improvements are not significant, then the value of the new method could be slightly lower).

(4) The proposed method is very simple and shows no significant novelty.

(5) The Bidirectional KL Divergence loss is a simple loss function and a natural choice for symmetry. However, the authors spent a considerable amount of space saying “we employed a BKLD loss for knowledge interaction” and the details about this loss function, which is unnecessary and may lead beginning readers to mistakenly believe that the paper proposes a novel loss function for fine-tuning knowledge interaction. Despite the clarity, I think some contents in this article are still redundant.

(6) Questions：

1) In eq 5, [CLS] is augmented by position embedding but [GIST] is not augmented. Why and would it make a difference if we augmented [GIST] with a specific position or what if we don’t augment [CLS]?

2) Why does the very first reference begin with [42]?

3) What if we change the value of μ -- the strength of aggregating Task-Specific Knowledge into [GIST]?

**Suitability:**

2

---

### Official Review · Reviewer_9D9a · 2024-05-21

**Rating:** 5
**Confidence:** 4

**Summary:**

This paper proposes a fusion framework of task-independent knowledge and specific knowledge, using lightweight learnable GIST token and bidirectional KL distance loss constrained learning process. The framework is plug-and-play, and the GIST is added to different methods to achieve performance improvements with a small increase in the number of parameters. Experiments on image classification, Few-shot classification, language understanding and visual-language tasks are carried out to prove the effectiveness and universality of the proposed method.

**Strengths:**

+ The paper is well-written and easy to understand.
+ The experimental section of this article is very complete and substantial, fully demonstrating the effectiveness and generalization of the framework on different tasks.
+ The framework proposed in this article is plug-and-play, which can be effectively used for different methods from different fields such as Adapter, LoRA, MaPLe, etc., and has strong universality.

**Limitations:**

- The roles of Gist classification loss and BKL loss are not clear, but BKL loss seems to play a greater role in ablation experiments. I would like to know why approximating the predicted distribution of CLS tokens and GIST tokens play such a big role?
- Does the inference model only use CLS Token predictions, or is it a aggregation of both CLS Token and GIST Token? Can you include a description of the reasoning process in your paper?
- In Fig 2 and 3, it seems inappropriate to describe the CLS Token at the output layer as frozen. First, it does not contain learnable parameters, but the aggregation of the weights of the previous layer. Second, the CLS Token also contains the weight of the learnable GIST Token.
- Does BKL loss bring more improvement than simple KL loss? Can you provide a comparison of your BKL losses to unidirectional KL losses?
- Why the the MaPLe's results in Table 4 are significantly worse than that in MaPLe [1]?
- The recently proposed DePT [2] is a plug-and-play and SOTA PEFT method for V-L models. Therefore, it is necessary to compare the performance gains of DePT and the proposed method on the same baseline method, e.g. on MaPLe.

[1] MaPLe: Multi-modal Prompt Learning, CVPR2023\
[2] DePT: Decoupled Prompt Tuning, CVPR2024

If all the above concerns are well addressed during the rebuttal phase, I will consider to give a clear ACCEPT.

**Suitability:**

3

---

### Official Review · Reviewer_to5r · 2024-05-27

**Rating:** 4
**Confidence:** 2

**Summary:**

This paper addresses the limitations of existing PEFT methods which do not adequately consider the interaction between task-agnostic and task-specific knowledge. The authors introduce a new framework, named GIST, which incorporates a learnable token called "Gist token" to enhance the connection between trainable parameters and downstream tasks, alongside a Bidirectional Kullback-Leibler Divergence (BKLD) loss for better knowledge interaction. The framework is tested across 35 datasets showing significant improvements in various tasks, including image classification and language processing, with minimal increases in parameter count.

**Strengths:**

1. The framework is extensively validated across a wide range of datasets and tasks
2. The GIST framework improves performance with only a slight increase in the number of parameters.

**Limitations:**

1. The use of a single Gist token to aggregate task-specific knowledge might lead to overfitting or limit the framework’s ability to generalize across highly diverse tasks.
2. The performance enhancements are partly attributed to the BKLD loss function, which may not be as effective if the assumptions about knowledge distribution do not hold.
3. It remain unclear the potential impacts of the Gist token's length and the temperature parameter in the BKLD loss on the framework's performance across different types of tasks.

**Suitability:**

3

---

### Meta-Review · Area_Chair_2FB2 · 2024-06-27

[review text omitted: it was posted to a different submission]